# Impact of repeat flooding on mental health and health-related quality of life: a cross-sectional analysis of the English National Study of Flooding and Health

Clare E French ![ORCID] ,[1,2] Thomas D Waite,[3] Ben Armstrong,[4] G. James Rubin,[5] English National Study of Flooding and Health Study Group, Charles R Beck,[1,2,5,6] Isabel Oliver[1,2,5,6]

Part of this work was presented at the Public Heath England Public Health Research and Science Conference 2018, Warwick, UK.

For numbered affiliations see end of article.

**Correspondence to**
Dr Clare E French;
clare.french@bristol.ac.uk

## ABSTRACT

**Objective** To assess the association between flooding/repeat flooding and: (1) psychological morbidity (anxiety, depression, post-traumatic stress disorder (PTSD)) and (2) health-related quality of life (HRQoL) at 6 months post-flooding.

**Design** Cross-sectional analysis of data from the English National Study of Flooding and Health.

**Setting** Cumbria, England.

**Participants** Questionnaires were sent to 2500 residential addresses at 6 months post-flooding; 590 people responded.

**Outcomes** Probable depression was assessed using the Patient Health Questionnaire, probable anxiety using the Generalised Anxiety Disorder scale and probable PTSD using the short-form PTSD checklist (PCL-6). HRQoL was assessed using the EQ-5D-5L. Mental health outcomes were analysed using logistic regression; HRQoL dimensions using ordinal regression; and summary index/Visual Analogue Scale scores using linear regression.

**Results** One hundred and nineteen participants had been flooded, over half of whom were experiencing a repeat flooding event (54%; n=64). Mental health outcomes were elevated among flooded compared with unaffected participants (adjusted OR for probable depression: 7.77, 95% CI: 1.51 to 40.13; anxiety: 4.16, 95% CI: 1.18 to 14.70; PTSD: 14.41, 95% CI: 3.91 to 53.13). The prevalence of depression was higher among repeat compared with single flooded participants, but this was not significant after adjustment. There was no difference in levels of anxiety or PTSD. Compared with unaffected participants, those flooded had lower EQ-5D-5L index scores (adjusted coefficient: −0.06, 95% CI: −0.12 to −0.01) and lower self-rated health scores (adjusted coefficient: −6.99, 95% CI: −11.96 to −2.02). There was, however, little difference in HRQoL overall between repeat and single flooded participants.

**Conclusions** Interventions are needed to help minimise the impact of flooding on people's mental health and HRQoL.

## INTRODUCTION

Flooding can have a range of effects on people's health, both physical and psychological.[1–5] Given projections that both the frequency

## Strengths and limitations of this study

► There is currently very little research exploring the impact of repeat flooding on mental health or health-related quality of life—our research addresses a gap in the evidence.
► We use data from the English National Study of Flooding and Health—a large, robust study with proven methodology.
► Three key mental health outcomes (depression, anxiety, post-traumatic stress disorder) as well as health-related quality of life were assessed, all using validated instruments.
► The tools used to assess psychological outcomes are indicative of probable diagnoses rather than being clinically diagnostic.
► Information was not available on the number or timing previous flooding events.

and severity of flooding events will continue to increase in the future,[6–9] populations living in flood-susceptible areas may be exposed to multiple flooding events. It is important that the impact of flooding, and repeat flooding, on people's health and well-being is thoroughly understood. This will help inform public health actions to mitigate adverse health consequences of future flooding events.

Beyond the acute health consequences of flooding, such as injuries and infections, the longer-term health consequences are thought to be primarily psychological.[6] The English National Study of Flooding and Health began in 2015 to assess the impact of widespread flooding in southern England. The study has assessed the impact of flooding and related disruption on mental health and well-being over several years. Data obtained 1 year after flooding in southern England showed higher levels of depression, anxiety and post-traumatic stress disorder (PTSD) in participants whose

homes had been flooded, and to a lesser extent, those whose lives had been disrupted by flooding compared with those who were unaffected.[10] Follow-up of participants at 2 years post-flooding demonstrated that mental health consequences had waned but were still noticeably higher in affected groups.[11] The impact of flooding on health-related quality of life (HRQoL) has not yet been reported on.

Several factors are known to be associated with the likelihood of experiencing adverse mental health outcomes post-flooding. These include sociodemographic characteristics such as age, gender, socioeconomic and education statuses, and factors related to the nature of flooding (eg, depth and duration).[12] Secondary stressors, including problems with relationships, loss of sentimental items and difficulties with insurance or compensation have been associated with poor psychological outcomes.[13 14] Meanwhile, societal-level factors such as social cohesion and support, as well as individual-level factors such as positive/proactive coping strategies can be protective against the development of mental illness in the aftermath of flooding.[2 15] Although there is less published research on the association between flooding and well-being/quality of life, the available evidence indicates that effects similarly occur through a complex set of inter-related pathways.[16]

The UK Climate Change Risk Assessment 2017 highlighted flooding as a risk to people, communities and buildings that requires more action.[17] More research is required to assess and understand how these risks are best tackled. There is currently very limited research exploring the impact of experiencing repeated flooding events on mental health and/or HRQoL, and the evidence to-date is inconclusive.[15 18–20]

In early December 2015, exceptionally heavy and persistent rainfall associated with storm 'Desmond', resulted in widespread flooding in Cumbria and other parts of northern Britain. Thousands of homes and businesses across Cumbria were flooded, and tens of thousands left without power.[21 22] Cumbria is an area that has been repeatedly affected by flooding—the region had experienced severe flooding just 6 years previously in 2009, and also in 2005.[22]

The objective of this study is to assess the association between flooding/repeat flooding and: (1) psychological morbidity (probable anxiety, probable depression, probable PTSD) and (2) HRQoL, in the Cumbria region at 6 months post-flooding.

## METHODS
### Study conduct
Details on the design and conduct of the English National Study of Flooding and Health have been published previously.[10] Recruitment and data collection in Cumbria largely mirrored the original survey conducted elsewhere. Recruitment packs (including consent forms, study information sheets and questionnaires) were sent to 2500 residential households in postcode areas known to have been affected by the 2015/16 floods within four

local authority areas (Allerdale, Carlisle, Eden, South Lakeland). The sample size was determined with the aim of sampling approximately 50% of flooded households. All adults in each household were invited to participate. Packs were sent out approximately 6 months post-flooding (May 2016), with two rounds of reminder letters sent to non-responders. Invitations were sent by post and participants had the option to respond either by post or online. Questionnaires were similar to those used for the original survey, with some minor clarification to the wording of questions, and the addition of the EQ-5D-5L instrument to assess HRQoL.

Participants were asked about their exposure to the 2015/2016 floods, and whether they had been previously flooded. They were categorised as:

▶ Unaffected—those who experienced no flooding or disruption during the 2015/2016 floods.
▶ Disrupted—those who did not have floodwater in liveable areas of their homes but reported experiencing disruption due to the floods (eg, floodwater in non-liveable areas such as a garage, disruption to loss of utilities, loss of access to services).
▶ Flooded—those who reported having floodwater in one or more liveable rooms of their home (eg, living room, kitchen, bedroom). For analyses of repeat flooding this group was further sub-categorised into:
  Repeat flooded—those who reported having been previously flooded (either in the same or a previous home).
  Single flooded—those who reported not having been previously flooded.

Demographic information collected included date of birth, sex, ethnicity, long-term illness or disability, education, employment and marital statuses. Postcodes of residence were used to assign Lower Super Output Area deprivation scores based on the Index of Multiple Deprivation.[23] These were ranked according to scores for all of England and categorised into quintiles.

Mental health outcomes were assessed using validated instruments—probable depression using the Patient Health Questionnaire (PHQ-2) depression sub-scale; probable anxiety using the Generalised Anxiety Disorder scale (GAD-2); and probable PTSD using the short-form PTSD checklist (PCL-6). Cut-off scores, used to indicate the likely presence of the condition, were ≥3 for PHQ-2/GAD-2 and ≥14 for PCL-6.[24–26] We also generated a combined variable indicating the presence of 'any' of these probable conditions.

HRQoL was assessed using the EQ-5D-5L. This is a validated two-part instrument. The first part is a descriptive system assessing five health-related dimensions (mobility, self-care, usual activities, pain/discomfort and anxiety/depression) on a five-point scale (no problems, slight problems, moderate problems, severe problems and extreme problems). Information on these five health states was converted into a single index value. The second part is a Visual Analogue Scale (VAS) which records the patient's self-rated health on a vertical scale marked from

0 to 100 where the endpoints are 'The best health you can imagine' and 'The worst health you can imagine'.[27]

SelectSurvey (ClassApps, USA) was used to develop the online questionnaire and capture online responses. Data returned on paper forms were double-entered using EpiData (EpiData Association, Denmark) and cross-checked to identify any errors.

## Data analysis

Data were analysed in STATA V.13.1 (StataCorp, USA). Participants who did not provide enough information to enable an exposure status to be assigned to them were excluded from analyses by exposure group. For analyses of mental health outcomes, participants who did not complete an outcome instrument were excluded from analyses of that outcome only. HRQoL analyses were restricted to those with complete information on both parts of the tool.

Mental health outcomes were analysed using binary logistic regression. For analyses of HRQoL, the EQ-5D-5L descriptive system ordered categorical variables were analysed using ordinal logistic regression. The proportional odds assumption was checked for each outcome using approximate likelihood-ratio tests. Continuous variables (EQ-5D-5L index score and VAS) both had left-skewed distributions and are thus presented as medians and IQRs. Medians were compared using the Mann-Whitney U test and adjusted analyses were conducted using linear regression with robust standard errors (SEs) to allow for the skewed distribution of the dependent variable.

Potential sociodemographic confounders of the association between flooding and mental health outcomes have been previously identified (age group, sex, local authority, ethnicity, marital status, education level, employment, deprivation score and pre-existing illness).[10] All (except ethnicity, due to very small numbers in the non-white ethnic group) were adjusted for a priori in all multivariable analyses for consistency and comparability between models.

## Patient and public involvement

A community flooding action group was involved in the design of the study. Volunteers identified by the National Flood Forum piloted early versions of the questionnaires, and the National Flood Forum Chief Executive is a member of our stakeholder group. Participants were invited to join a study group to support the project and keep track of its progress. Study findings are being disseminated to participants via reports published online.

## RESULTS
## Response rates

Completed questionnaires were received from 590 participants (household response rate: 24.4%). More than one individual responded from 19 households giving a total of 628 participating individuals. Most participants responded using the paper questionnaire, with 7.0% (n=44) responding online. Subsequent analyses were conducted on the 531 participants who provided sufficient information to enable an exposure status to be assigned to them.

## Participant characteristics

The sociodemographic characteristics of study participants are provided in online supplementary appendix I. Overall, the median age was 62 years (IQR: 49–72 years), 58.8% were female, 99.6% were of white ethnicity, 51.6% lived in areas belonging to the two least deprived quintiles and 76.5% had a long-term illness or disability.

## Exposure to flooding and psychological morbidity

Overall, 16.9% of participants were unaffected by the floods, 60.6% were disrupted and 22.4% were flooded.

After adjusting for sociodemographic factors, there was an elevated odds of each mental health outcome among flooded participants compared with those unaffected (adjusted OR (aOR) for probable depression: 7.77, 95% CI (CI): 1.51 to 40.13; anxiety: 4.16, 95% CI: 1.18 to 14.70; PTSD: 14.41, 95% CI: 3.91 to 53.13; and for 'any' of these mental health outcomes: 11.81, 95% CI: 3.99 to 34.96). The adjusted odds of probable depression, PTSD and any outcome (but not anxiety) were also somewhat raised for those who experienced disruption due to flooding but not significantly so (table 1).

Of those flooded during the 2015/2016 floods, over half (54%; 64/119) had been previously flooded (60 at the same home and 4 at a previous home). The prevalence of probable depression was elevated among those exposed to repeated, as compared with a single flooding event, but the difference reduced on adjustment for potential confounders, and was not significant. For probable anxiety and PTSD, there was little difference in prevalence among people who experienced single and repeat flooding, either before or after adjustment (table 1).

## Exposure to flooding and HRQoL

Most participants (97.9%, n=520) had complete information on both parts of the EQ-5D-5L.

### Index scores

Median EQ-5D-5L index scores are presented in table 2. In adjusted analyses, flooded participants (all) had lower EQ-5D-5L index scores than unaffected participants (adjusted coefficient: −0.06, 95% CI: −0.12 to −0.01). There was little evidence that scores differed between repeat and single flooded participants.

### Health-related dimensions

The level of problems reported by participants on each dimension, according to exposure to flooding are provided in online supplementary appendix II.

Flooded participants were more likely to have higher anxiety/depression scores than those unaffected (aOR: 8.59, 95% CI: 4.02 to 18.36; table 3), also more self-care and activity problems, though not significantly so. There was little evidence that the flooded group had more

**Table 1** Prevalence and ORs (crude and adjusted) of mental health outcomes according to exposure to flooding

| | n/N (%)* | OR (95% CI) | aOR† ‡ (95% CI) |
|---|---|---|---|
| **Model 1: Comparing flooded and disrupted with unaffected participants** | | | |
| Probable depression | | | |
| Unaffected | 2/83 (2.4) | Ref | Ref |
| Disrupted | 15/292 (5.1) | 2.19 (0.49 to 9.80) | 1.38 (0.27 to 6.96) |
| Flooded | 25/106 (23.6) | 12.50 (2.87 to 54.52) | 7.77 (1.51 to 40.13) |
| Probable anxiety | | | |
| Unaffected | 5/83 (6.0) | Ref | Ref |
| Disrupted | 19/289 (6.6) | 1.10 (0.40 to 3.03) | 0.91 (0.27 to 3.08) |
| Flooded | 24/107 (22.4) | 4.51 (1.64 to 12.41) | 4.16 (1.18 to 14.70) |
| Probable PTSD | | | |
| Unaffected | 3/87 (3.4) | Ref | Ref |
| Disrupted | 30/300 (10.0) | 3.11 (0.93 to 10.45) | 2.05 (0.57 to 7.40) |
| Flooded | 49/112 (43.8) | 21.78 (6.49 to 73.08) | 14.41 (3.91 to 53.13) |
| 'Any' probable mental health outcome | | | |
| Unaffected | 6/82 (7.3) | Ref | Ref |
| Disrupted | 41/285 (14.4) | 2.13 (0.87 to 5.21) | 1.88 (0.67 to 5.32) |
| Flooded | 52/107 (48.6) | 11.98 (4.08 to 29.86) | 11.81 (3.99 to 34.96) |
| **Model 2: Comparing repeat with single flooded participants** | | | |
| Probable depression | | | |
| Single flooded | 8/49 (16.3) | Ref | Ref |
| Repeat flooded | 17/57 (29.8) | 2.18 (0.85 to 5.61) | 1.55 (0.44 to 5.46) |
| Probable anxiety | | | |
| Single flooded | 11/50 (22.0) | Ref | Ref |
| Repeat flooded | 13/57 (22.8) | 1.05 (0.42 to 2.61) | 0.89 (0.26 to 2.97) |
| Probable PTSD | | | |
| Single flooded | 21/51 (41.2) | Ref | Ref |
| Repeat flooded | 28/61 (45.9) | 1.21 (0.57 to 2.57) | 0.73 (0.27 to 1.97) |
| 'Any' probable mental health outcome | | | |
| Single flooded | 22/48 (45.8) | Ref | Ref |
| Repeat flooded | 30/59 (50.8) | 1.22 (0.57 to 2.62) | 1.03 (0.39 to 2.73) |

*N is the total number of participants in each exposure group for whom a complete answer to this outcome instrument was provided, and % is the proportion of those who have the relevant (probable) condition.
†Adjusted for age group, sex, local authority, marital status, education level, employment, deprivation score and pre-existing illness.
‡Number of observations included in the multivariable models were: probable depression n=446, probable anxiety n=444, probable PTSD n=464, 'any' probable mental health outcome n=440. Both models included the same participants but the reference group differs. ORs are only presented for the comparison of interest.
aOR, adjusted OR; PTSD, post-traumatic stress disorder.

mobility problems or pain/discomfort than those unaffected. There was little evidence of a difference between repeat and single flooded participants on any dimension (table 3). There was no evidence against the proportional odds assumption for any of the five dimensions.

### VAS scores
Median VAS scores are presented in table 4. Scores were 78 (IQR 69 to 90) for flooded participants, notably lower than for those unaffected (90, IQR: 75 to 93). This difference remained after adjustment (adjusted coefficient of VAS scores among flooded vs unaffected participants: −6.99, 95% CI: −11.96 to −2.02). There was little evidence of a difference in VAS scores between the repeat and single flooded groups. For analyses of both index scores and VAS scores the key assumptions of linear regression were met; robust SEs were used which dealt with the non-normality of the data.

### DISCUSSION
This study provides strong evidence of an association between flooding, be it a single or repeat episode, and

**Table 2** Univariable and multivariable analyses of the association between exposure to flooding and EQ-5D-5L index values

| | Exposure group | Median (IQR) index value | Crude coefficient (95% CI)* | Adjusted coefficient (95% CI)* † |
|---|---|---|---|---|
| Model 1 | Unaffected | 1.00 (0.77 to 1.00) | Ref | Ref |
| | Disrupted | 1.00 (0.77 to 1.00) | −0.00 (−0.52 to 0.04) | −0.00 (−0.44 to 0.04) |
| | Flooded | 0.85 (0.69 to 1.00) | −0.08 (−0.14 to −0.02) | −0.06 (−0.12 to −0.01) |
| Model 2 | Single flooded | 0.84 (0.68 to 0.88) | Ref | Ref |
| | Repeat flooded | 0.85 (0.69 to 1.00) | 0.00 (−0.08 to 0.08) | 0.02 (−0.05 to 0.09) |

*CIs estimated using robust SEs.
†Adjusted for age group, sex, local authority, marital status, education level, employment, deprivation score and pre-existing illness.

probable mental health outcomes. Flooded participants also tended to have poorer HRQoL than those unaffected by the floods. Analyses of the dimension-specific descriptive EQ-5D-5L system indicate that anxiety/depression may be an important driver of this association. There was little evidence of a difference in either probable mental health outcomes or HRQoL between repeat and single flooded participants, although data were suggestive of a greater level of depression in those repeatedly flooded.

The UK Climate Change Risk Assessment 2017 identifies flooding as a major risk to people and the built environment; more action is needed to mitigate this risk.[17] The UK government has committed to actions, such as ensuring that people have good access to information to assess the risks, including health risks, of flooding; helping people to make their properties more resistant to flooding; and ensuring plans are in place to predict and respond to flooding incidents.[28] Our research should be used to inform that government action—although we focus on the impact of flooding on health and well-being, the evidence presented here supports the case for good town planning, flood and coastal risk management, and the implementation of the 25-year Environment Plan.[28]

The overall associations between being flooded (vs unaffected) and mental health outcomes were consistent with those measured in the cohort affected by the floods in the winter of 2013/2014 in southern England.[10] Those floods largely affected affluent areas potentially limiting the generalisability of the findings beyond that population. Participants from Cumbria resided in areas of comparatively greater deprivation; our findings thus demonstrate that the increased risks of mental health outcomes following flooding also pertain to those living in more deprived areas. This is consistent with low income having been previously reported as a known driver of vulnerability to the effects of flooding.[29] Of note, a strong association between flooding and anxiety/depression was found using the EQ-5D-5L, corroborating findings obtained using mental health outcome assessment tools.

There is limited previous research on the impact of flooding on HRQoL with which to draw comparisons, and we are not aware of any previously published comparable data on the impact of repeat flooding specifically. A mixed-methods study in two UK regions found that flooding had a marked impact on well-being. This occurred via a complex and diverse range of interacting factors acting at both the individual and societal level.[16] Data from China showed poorer quality of life among those affected by floods compared with those unaffected, but the EQ-5D-5L tool was not used and the findings are not directly comparable.[30]

That over half of flooded participants in our sample had experienced repeat flooding is striking. It may be hypothesised that this group will suffer greater mental health and HRQoL effects than those experiencing single event due to the cumulative effects of these experiences. However, our research indicates that any association may be more nuanced. The data were suggestive of higher levels of depression, but not other mental health outcomes or overall HRQoL, among repeat compared with single flooded participants. Of note, the number of events were small in some groups resulting in imprecise estimates—the data should thus be interpreted with some caution. The lack of association found in our study does, however, concur with a recently published Australian study which reported a general lack of association between exposure to repeat natural disasters and the broader psychiatric disorder spectrum.[31]

Repeat flooding may impact on mental health in a variety of ways, though the evidence is inconclusive.[15 18 19] At the individual level, experiencing a repeat episode of flooding may trigger memories of previous flooding episodes, intensifying the distress experienced.[18] Those who have been previously flooded may also experience anticipatory anxiety and stress about the prospect of future flooding.[32 33] Conversely, people's individual and collective coping capacity may be increased, and they may display greater resilience to future flooding events.[34] There is also evidence to suggest that previous exposure to disasters may have an 'inoculation' effect, particularly among older adults (the median age of our study population was 62 years).[20 35 36] Meanwhile, factors acting at the community level may mitigate the risks of adverse consequences of future floods through the development or strengthening of community networks and increased social cohesion.[16 32 37] Finally, past flood events may have resulted in improvements to infrastructure, flood defences and response systems, reducing the extent of subsequent floods and the impact on people's lives.

General limitations of the National Study of Flooding and Health have been outlined elsewhere.[10] Our questionnaire response rate of 24% is in line with other similar

**Table 3** Univariable and multivariable analyses of EQ-5D-5L health-related dimensions according to exposure to flooding

| | OR (95% CI) | aOR (95% CI)* † |
|---|---|---|
| **Model 1: Comparing flooded and disrupted with unaffected participants** | | |
| Mobility problems | | |
| Unaffected | Ref | Ref |
| Disrupted | 0.87 (0.51 to 1.49) | 0.98 (0.48 to 2.02) |
| Flooded | 1.09 (0.59 to 2.02) | 1.04 (0.44 to 2.46) |
| Self-care problems‡ | | |
| Unaffected | Ref | Ref |
| Disrupted | 1.73 (0.58 to 5.12) | 1.61 (0.41 to 6.31) |
| Flooded | 1.98 (0.60 to 6.52) | 1.62 (0.33 to 7.91) |
| Activity problems | | |
| Unaffected | Ref | Ref |
| Disrupted | 1.15 (0.64 to 2.07) | 1.59 (0.70 to 3.60) |
| Flooded | 1.68 (0.87 to 3.25) | 1.85 (0.72 to 4.79) |
| Pain/discomfort | | |
| Unaffected | Ref | Ref |
| Disrupted | 0.92 (0.58 to 1.46) | 0.95 (0.54 to 1.68) |
| Flooded | 1.14 (0.67 to 1.95) | 1.01 (0.50 to 2.02) |
| Anxiety/depression | | |
| Unaffected | Ref | Ref |
| Disrupted | 1.51 (0.84 to 2.70) | 1.46 (0.73 to 2.91) |
| Flooded | 7.72 (4.11 to 14.50) | 8.59 (4.02 to 18.36) |
| **Model 2: Comparing repeat with single flooded participants** | | |
| Mobility problems | | |
| Single flooded | Ref | Ref |
| Repeat flooded | 1.16 (0.52 to 2.63) | 0.85 (0.28 to 2.54) |
| Self-care problems‡ | | |
| Single flooded | Ref | Ref |
| Repeat flooded | 0.82 (0.23 to 3.01) | 0.42 (0.08 to 2.29) |
| Activity problems | | |
| Single flooded | Ref | Ref |
| Repeat flooded | 0.74 (0.33 to 1.64) | 0.71 (0.24 to 2.11) |
| Pain/discomfort | | |
| Single flooded | Ref | Ref |
| Repeat flooded | 0.88 (0.43 to 1.77) | 0.52 (0.21 to 1.26) |
| Anxiety/depression | | |
| Single flooded | Ref | Ref |
| Repeat flooded | 1.20 (0.62 to 2.32) | 1.13 (0.51 to 2.51) |

*Adjusted for age group, sex, local authority, marital status, education level, employment, deprivation score and pre-existing illness.
†Number of observations included in all multivariable models was 484. Both models 1 and 2 included the same participants, but the reference group differed. ORs are only presented for the comparison of interest.
‡Due to very small numbers of participants with 'severe/extreme problems' on the self-care dimension, the 'severe/extreme' and 'moderate' categories were combined.
aOR, adjusted OR.

postal surveys.[38 39] Although this relatively low response rate is a potential limitation of our study, it would only bias the observed associations between flooding and mental health/HRQoL if response was differential with respect to both flooding and health outcomes. Although this is a possibility, we do not have evidence to suggest that any such differential response would be substantial. Another limitation of these analyses specifically is that we do not have any information on the number, timing or extent of previous flooding experiences. Such information would be helpful in interpreting our findings and should be considered when designing future research studies on the health impacts of flooding.

Though validated instruments, the tools used to assess psychological outcomes are not clinically diagnostic. Our data therefore indicate 'probable' diagnoses which are likely an overestimate of diagnosable mental health conditions. Meanwhile, the EQ-5D-5L is a validated and widely used tool that has been previously used in UK populations.[40] The tool was well completed (98% of participants had complete data) indicating its acceptability in our study. However, some of the dimensions assessed may not be directly relevant to the types of illnesses that may occur as a consequence of flooding. At the same time, due to space constraints on the questionnaire, we were unable to assess all psychological variables that are of theoretical interest following repeat flooding. For example, anger might be elevated in flooded, and especially repeat flooded, participants and may be worthy of future investigation.

Our findings may not be generalisable to all populations affected by flooding (eg, there were few participants of non-white ethnicity). However, all comparison groups were drawn from the same population and key sociodemographic confounding factors were adjusted for in the analyses.

## CONCLUSIONS AND RECOMMENDATIONS

Flooding can have a profound effect on people's psychological health, and aspects of their HRQoL. Though we found little evidence that people exposed to repeat flooding were at increased risk of adverse outcomes compared with those exposed to a single event, it should be remembered that these individuals are at risk of experiencing adverse outcomes each time they experience flooding.

That over half of participants in this study were experiencing a repeat flooding event highlights the need for strengthening flood defences and mitigating risks of future flooding in areas at risk or previously affected. The UK's National Adaptation Programme for climate change recommends actions for reducing flood risk and harm from flooding including effective land use planning[28]; the evaluation of these planned actions should include an assessment of their impact on people's mental health and well-being.

Interventions are needed to help minimise the impact of flooding on mental health and well-being. These may include early warning systems which can reduce the effects of flooding on mental health,[41] and community-based

**Table 4**  Univariable and multivariable analyses of the association between exposure to flooding and EQ-5D-5L VAS scores

| | Exposure group | Median (IQR) VAS score | Crude coefficient (95% CI)* | Adjusted coefficient (95% CI)* † |
|---|---|---|---|---|
| Model 1 | Unaffected | 90 (75 to 93) | Ref | Ref |
| | Disrupted | 85 (75 to 90) | −2.18 (−6.15 to 1.79) | −1.77 (−5.45 to 1.91) |
| | Flooded | 78 (69 to 90) | −8.66 (−13.66 to −3.67) | −6.99 (−11.96 to −2.02) |
| Model 2 | Single flooded | 75 (70 to 85) | Ref | Ref |
| | Repeat flooded | 80 (65 to 90) | −0.64 (−7.78 to 6.50) | 0.76 (−7.06 to 8.60) |

*CIs estimated using robust SEs.
†Adjusted for age group, sex, local authority, marital status, education level, employment, deprivation score and pre-existing illness.

psychosocial support such as psychological first aid in the immediate aftermath of flood events. Further research on the effectiveness of these and other interventions to reduce the impact of flooding on people's health would be informative in shaping future intervention strategies.

Providing people with other approaches to reduce their susceptibility to the effects of flooding, such as efforts to increase community cohesion and develop support networks in preparedness for potential future flooding events, may also be beneficial. Finally, ensuring that adequate and effective mental health services are available for community members following flooding is vital.

**Author affiliations**
[1]NIHR Health Protection Unit in Evaluation of Interventions, Population Health Sciences, Bristol Medical School, Bristol, UK
[2]Population Health Sciences, Bristol Medical School, University of Bristol, Bristol, UK
[3]National Infection Service, Public Health England, London, UK
[4]NIHR Health Protection Research Unit in Environmental Change and Health at the London School of Hygiene and Tropical Medicine, London, UK
[5]Department of Psychological Medicine, Weston Education Centre, King's College London NIHR Health Protection Research Unit in Emergency Preparedness and Response, London, UK
[6]Field Service South West, National Infection Service, Public Health England, Bristol, UK

**Acknowledgements**  We would like to thank the participants in this study and are grateful for the support we received from Colin Cox, Director of Public Health Cumbria County Council and the North West Public Health England Centre. We acknowledge the role of other members of the English National Study of Flooding and Health Study Group in the design of the National Study of Flooding and Health: Katerina Chaintarli, Angie Bone, Richard Amlôt, Sari Kovats, Mark Reacher and Giovanni Leonardi. We would also like to thank all PHE staff who contributed to study administration, set up and data entry.

**Contributors**  CEF led the design and conduct of these analyses, with expert statistical advice from BA, and drafted the manuscript. English National Study of Flooding and Health Study Group members including named authors IO, TDW, BA, CRB and GJR designed the National Study of Flooding and Health and acquired the data. IO, TDW, BA, CRB and GJR contributed to the design of the analyses and the interpretation the findings. All authors contributed to the writing of the manuscript and approved it for submission.

**Funding**  The research was funded in part by the National Institute for Health Research Health Protection Research Units (NIHR HPRU) in Emergency Preparedness and Response at King's College London, Environmental Change at the London School of Hygiene and Tropical Medicine and Evaluation of Interventions at the University of Bristol, in partnership with Public Health England (PHE). The views expressed are those of the authors and not those of the NHS, the NIHR, The Department of Health or Public Health England.

**Competing interests**  None declared.

**Patient consent for publication**  Not required.

**Ethics approval**  Ethical approval for the study was granted by the Psychiatry, Nursing and Midwifery Research Ethics Subcommittee at King's College London [Reference PNM 1314 152]. All participants consented to participate in the study.

**Provenance and peer review**  Not commissioned; externally peer reviewed.

**Data availability statement**  The datasets analysed in this study are available from Public Health England Field Epidemiology Service on reasonable request.

**ORCID iD**
Clare E French http://orcid.org/0000-0002-6943-7353

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
