## [Reviewer comments · BMJ Open]

ARTICLE DETAILS

TITLE (PROVISIONAL)	Impact of repeat flooding on mental health and health-related quality of life: a cross-sectional analysis of the English National Study of Flooding and Health
AUTHORS	French, Clare E; Waite, Thomas; Armstrong, Ben; Rubin, GJ; National Study of Flooding and Health Study Group, English; Beck, Charles; Oliver, Isabel

VERSION 1 – REVIEW

REVIEWER	Gerry FitzGerald Queensland University of Technology, Queensland Australia
REVIEW RETURNED	30-May-2019

GENERAL COMMENTS	The research is well constructed and the paper presented in a manner appropriate for publication. My only minor suggestion would be to mention the potential selection bias. As I read it, surveys were sent to 2,500 households identified by affected postcodes resulting in a 22.4%. There may be some degree of selection bias which could be mentioned in the limitations. The findings are important and the authors may wish to speculate on potential community based therapeutic interventions (psychological first aid) as well as the land use planning. I hope this is a longitudinal study and that long term follow up is intended. The international evidence suggests that short term mental health impacts are indeed common. However, there is emerging evidence that long term consequences are also common and may relate less directly to the event itself but rather to the poorly handled recovery phase. Put simply, people can handle their house being flooded but it is the three year fight with the insurance companies that causes considerable distress. I would be most interested in the long term consequences and the factors that impact on those consequences. It may be difficult to prevent flooding in the first instance but it is possible to better manage the post event recovery.
---

REVIEWER	Lennart Reifels Centre for Mental Health Melbourne School of Population and Global Health The University of Melbourne Australia
REVIEW RETURNED	24-Jun-2019

GENERAL COMMENTS	The study makes a welcome contribution to the burgeoning literature on the mental health impacts of repeat disaster exposure. The methodology appears to be sound and existing study limitations have been outlined (i.e., cross-sectional design, relatively crude prior disaster exposure measure, no formal diagnoses, small number of
--

events in some groups resulting in imprecise estimates).

My relatively minor comments and suggestions to improve the manuscript are as follows:

RESULTS

- (p.7, first paragraph): reports a different median age (69 years) to Appendix I (62.4 years)

DISCUSSION

- The particular age demographic of the sample (median age 62.4 years, 76.5% long-term illness) could briefly be discussed to contextualise study findings.

- Realising that multivariate analyses controlled for demographic variables, I am still curious as to why women in this sample were more likely to be in flood-affected or disrupted groups, while men were more likely to be unaffected (see Appendix I). Could you speculate or explain? Given the increased likelihood of flood exposure among women, did you observe differential odds of adverse mental health outcomes of flood exposure for women and men? Along similar lines, were women or men more likely to experience repeat flood exposure?

- Another theory or stream of disaster literature that could potentially be worth considering in this context (in view of the specific age demographic of the sample) discusses a potential protective 'inoculation' effect of prior disaster exposure, particularly among the elderly.

Knight, B. G., Gatz, M., Heller, K., & Bengtson, V. L. (2000). Age and emotional response to the Northridge earthquake: A longitudinal analysis. *Psychology and Aging*, 15(4), 627–634.

Norris, F. H., & Murrell, S. A. (1988). Prior experience as a moderator of disaster impact on anxiety symptoms in older adults. *American Journal of Community Psychology*, 16(5), 665–683.

Shrira, A., Palgi, Y., Hamama-Raz, Y., Goodwin, R., & Ben-Ezra, M. (2014). Previous exposure to the World Trade Center terrorist attack and posttraumatic symptoms among older adults following Hurricane Sandy. *Psychiatry: Interpersonal & Biological Processes*, 77(4), 374–385.

- The somewhat counter-intuitive lack of an effect of repeat flood exposure on mental health in this study concurs with findings from previous cross-sectional studies regarding the impacts of repeat natural disaster exposure on the broader psychiatric disorder spectrum. By contrast, repeat natural disaster exposure did appear to be associated with suicidality.

Reifels L, Mills K, Dückers MLA, O'Donnell ML (2019). Psychiatric epidemiology and disaster exposure in Australia. *Epidemiology and Psychiatric Sciences*. 28(3), 310-320.

Reifels L, Spittal M, Dückers M, Mills K, Pirkis J (2018). Suicidality risk and (repeat) disaster exposure: Findings from a nationally representative population survey. *Psychiatry: Interpersonal & Biological Processes*. 81(2), 158-172.

	 • In view of the study focus on repeat flood exposure, the discussion could provide some further directions to advance future repeat disaster exposure research. TABLE 1  • Table title could specify “at six months” APPENDIX I  • Should report both n and % for Education level (Degree or above) - disrupted Minor Typos  • (p. 4, 3rd paragraph, last sentence): “... a complex set [of] interrelated pathways” • (p. 5, 4th paragraph, last sentence): “... a combined variable indic[aj]ting”
--	---

VERSION 1 – AUTHOR RESPONSE

Reviewer: 1

Reviewer Name: Gerry FitzGerald

Institution and Country: Queensland University of Technology, Queensland Australia

Please state any competing interests or state ‘None declared’: None apart from similar research and publications

The research is well constructed and the paper presented in a manner appropriate for publication.

My only minor suggestion would be to mention the potential selection bias. As I read it, surveys were sent to 2,500 households identified by affected postcodes resulting in a 22.4%. There may be some degree of selection bias which could be mentioned in the limitations.

Author response: As per our response to the Editor’s request, we have added this to the limitations of the Discussion, paragraph 7: “Our questionnaire response rate of 24% is in line with other similar postal surveys [Lamond et al, 2015; Paranjothy et al, 2011]. Although this relatively low response rate is a potential limitation of our study, it would only bias the observed associations between flooding and mental health / HRQoL if response was differential with respect to both flooding and health outcomes. Although this is a possibility, we don’t have evidence to suggest that any such differential response would be substantial.”

The findings are important and the authors may wish to speculate on potential community based therapeutic interventions (psychological first aid) as well as the land use planning.

Author response: We have highlighted the potential importance of both psychological first aid and effective land use planning to the ‘Conclusions and recommendations’ section (paragraphs 2 and 3): “.....The UK’s National Adaptation Programme for climate change recommends actions for reducing flood risk and harm from flooding including effective land use planning;[27] the evaluation of these planned actions should include an assessment of their impact on people’s mental health and wellbeing.”

Interventions are needed to help minimise the impact of flooding on people’s mental health and their wellbeing. These may include early warning systems which can reduce the effects of flooding on mental health, [35] and community-based therapeutic actions such as psychological first aid in the immediate aftermath of flood events....”

I hope this is a longitudinal study and that long term follow up is intended. The international evidence suggests that short term mental health impacts are indeed common. However, there is emerging evidence that long term consequences are also common and may relate less directly to the event itself but rather to the poorly handled recovery phase. Put simply, people can handle their house being flooded but it is the three-year fight with the insurance companies that causes considerable distress. I would be most interested in the long-term consequences and the factors that impact on

those consequences. It may be difficult to prevent flooding in the first instance but it is possible to better manage the post event recovery.

Author response: The English National Study of Flooding and Health is a longitudinal study. Two-year follow-up data from southern England has already been published: Jermacane D et al.

The English National Cohort Study of Flooding and Health: the change in the prevalence of psychological morbidity at year two. BMC Public Health. 2018 Mar 7;18(1):330, and an analysis of the three-year follow-up data for the same region has been submitted for publication.

The National Study of Flooding and Health team also recently published a paper on the effect of insurance-related factors on the association between flooding and probable mental health outcomes which may be of interest: Mulchandani R et al. Effect of Insurance-Related Factors on the Association between Flooding and Mental Health Outcomes. Int J Environ Res Public Health. 2019 Apr 2;16(7).

Reviewer: 2

Reviewer Name: Lennart Reifels

Institution and Country:

Centre for Mental Health

Melbourne School of Population and Global Health The University of Melbourne Australia

Please state any competing interests or state 'None declared': None declared

The study makes a welcome contribution to the burgeoning literature on the mental health impacts of repeat disaster exposure. The methodology appears to be sound and existing study limitations have been outlined (i.e., cross-sectional design, relatively crude prior disaster exposure measure, no formal diagnoses, small number of events in some groups resulting in imprecise estimates).

My relatively minor comments and suggestions to improve the manuscript are as follows:

RESULTS

- (p.7, first paragraph): reports a different median age (69 years) to Appendix I (62.4 years)
Author response: Thank you for picking up on this typo – we have corrected it in the text.

DISCUSSION

- The particular age demographic of the sample (median age 62.4 years, 76.5% long-term illness) could briefly be discussed to contextualise study findings.
Author response: We have integrated information on the particular age demographic into the following (newly added in response to one of your later suggestions) sentence: "There is also evidence to suggest that previous exposure to a disaster may have an 'inoculation' effects, particularly among older adults (the median age of our study population was 62 years) [Knight et al, 2000; Norris et al, 1988; Shira et al, 2014]"(Discussion, paragraph 6)

- Realising that multivariate analyses controlled for demographic variables, I am still curious as to why women in this sample were more likely to be in flood-affected or disrupted groups, while men were more likely to be unaffected (see Appendix I). Could you speculate or explain? Given the increased likelihood of flood exposure among women, did you observe differential odds of adverse mental health outcomes of flood exposure for women and men? Along similar lines, were women or men more likely to experience repeat flood exposure?

Author response: It seems unlikely that flood exposure is differential according to sex. The observed difference (as seen in Appendix I) is more likely a reflection of differences in who responded to our questionnaire i.e. perhaps women exposed to flooding are more likely to respond to the questionnaire than men who were exposed. In our adjusted analyses there was little difference in mental health outcomes in women as compared with men (no significant difference for depression or PTSD but a somewhat higher odds of anxiety), and there was no significant difference in health-related quality of life. We have not specifically reported on this in the text as the aim of the analyses was to assess the association between flooding and health outcomes after taking account of socio-demographic factors rather than to explore which socio-demographic factors are associated with poorer health outcomes in the aftermath of flooding events.

- Another theory or stream of disaster literature that could potentially be worth considering in this context (in view of the specific age demographic of the sample) discusses a potential protective 'inoculation' effect of prior disaster exposure, particularly among the elderly.

Knight, B. G., Gatz, M., Heller, K., & Bengtson, V. L. (2000). Age and emotional response to the Northridge earthquake: A longitudinal analysis. *Psychology and Aging*, 15(4), 627–634.

Norris, F. H., & Murrell, S. A. (1988). Prior experience as a moderator of disaster impact on anxiety symptoms in older adults. *American Journal of Community Psychology*, 16(5), 665–683.

Shrira, A., Palgi, Y., Hamama-Raz, Y., Goodwin, R., & Ben-Ezra, M. (2014). Previous exposure to the World Trade Center terrorist attack and posttraumatic symptoms among older adults following Hurricane Sandy. *Psychiatry: Interpersonal & Biological Processes*, 77(4), 374–385.

Author response: Thank you for providing these references. We have added this sentence to our discussion around the varying ways in which repeat flooding may (or may not) impact on mental health outcomes (Discussion, paragraph 6) "...Conversely, people's individual and collective coping capacity may be increased, and they may display greater resilience to future flooding events.[32] There is also evidence to suggest that previous exposure to a disaster may have an 'inoculation' effects, particularly among older adults (the median age of our study population was 62 years) [Knight et al, 2000; Norris et al, 1988; Shira et al, 2014]."

- The somewhat counter-intuitive lack of an effect of repeat flood exposure on mental health in this study concurs with findings from previous cross-sectional studies regarding the impacts of repeat natural disaster exposure on the broader psychiatric disorder spectrum. By contrast, repeat natural disaster exposure did appear to be associated with suicidality.

Reifels L, Mills K, Dückers MLA, O'Donnell ML (2019). Psychiatric epidemiology and disaster exposure in Australia. *Epidemiology and Psychiatric Sciences*. 28(3), 310-320.

Reifels L, Spittal M, Dückers M, Mills K, Pirkis J (2018). Suicidality risk and (repeat) disaster exposure: Findings from a nationally representative population survey. *Psychiatry: Interpersonal & Biological Processes*. 81(2), 158-172.

Author response: Thank you for these references. We have added the following text to the Discussion, paragraph 5 "...The lack of association in our study does, however, concur with a recently published Australian study which found a general lack of association between exposure to repeat natural disasters and the broader psychiatric disorder spectrum [Reifels et al, 2018]."

- In view of the study focus on repeat flood exposure, the discussion could provide some further directions to advance future repeat disaster exposure research.

Author response: In response to this suggestion we have now integrated some suggestions for future repeat flooding research into the discussion section, as follows:

Discussion, paragraph 7 (Study limitations): "...Another limitation of these analyses specifically is that we do not have any information on the number, timing or extent of previous flooding experiences. Such information would be helpful in interpreting our findings and should be considered when designing future research studies on the health impacts of flooding."

Conclusions and recommendations, paragraph 3: "Interventions are needed to help minimise the impact of flooding on people's mental health and their wellbeing. These may include early warning systems which can reduce the effects of flooding on mental health, [35] and community-based therapeutic actions (psychological first aid) in the immediate aftermath of flood events. Further research on the effectiveness of these and other interventions to reduce the impact of flooding on people's health would be informative in shaping future intervention strategies."

TABLE 1

- Table title could specify “at six months”

Author response: Since all tables (and indeed all analyses) report outcomes at six months post-flooding, and this is clearly stated both in the objectives (page 4, last paragraph) and in the Methods (first paragraph), if we add this additional piece of information to Table 1 we would need to do so for all other tables too. On balance, given that some of the table titles are already quite long, we decided it is probably better not to make this change.

APPENDIX I

- Should report both n and % for Education level (Degree or above) - disrupted

Author response: We have added this.

Minor Typos

- (p. 4, 3rd paragraph, last sentence): “... a complex set [of] interrelated pathways”
- (p. 5, 4th paragraph, last sentence): “... a combined variable indic[a]ting”

Author response: Thank you – we have corrected both typos.

VERSION 2 – REVIEW

REVIEWER	Gerry FitzGerald Queensland University of Technology, Brisbane Australia
REVIEW RETURNED	23-Aug-2019

GENERAL COMMENTS	The aim so this paper is to identify the six month incidence of mental health problems amongst people whose houses have been flooded; either once or repeatedly. As such the paper is appropriately structured around this aim and the findings lined to the study's objectives. I have two comments that the authors may wish to address: Firstly, the background recognises prior research by the group into 12mont and 2 year mental health issues following flooding. It is not entirely clear how this six months follow up study differentiates form these and form studies undertaken immediately after the flooding. The research question is does mental health sequalae following flooding change over time; either in severity or incidence. secondly this point prevalence survey approach does not provide an enriched understanding of the actors that may be influencing the associations. Similar studies elsewhere involving qualitative analysis suggest that the post flooding management may be a significant factors along with the reality of risk exposure. Put simply, people can handle the flooding but have difficulty fighting with the insurance companies over the compensation. Perhaps the authors may wish to discuss the possible pathophysiological rationale of their observations in a little more depth.
--

REVIEWER	Lennart Reifels Centre for Mental Health Melbourne School of Population and Global Health The University of Melbourne Australia
REVIEW RETURNED	08-Aug-2019

GENERAL COMMENTS	The authors have satisfactorily addressed all feedback points raised by this reviewer. Minor remaining typos can likely be addressed during copy-editing:
---

	(p.32, line): "The lack of association found [in] our study" (p.33, line 13): "may have an 'inoculation' effect[s]" (p.32, second paragraph): Since 'therapeutic interventions' are not usually indicated in the immediate aftermath of disasters, I would recommend to refer to Psychological First Aid as a "psychosocial support" strategy.
--	--

VERSION 2 – AUTHOR RESPONSE

Reviewer: 1

Reviewer Name: Gerry FitzGerald

Institution and Country: Queensland University of Technology, Brisbane Australia Please state any competing interests or state 'None declared': None except for similar publications in the field

Please leave your comments for the authors below

The aim so this paper is to identify the six month incidence of mental health problems amongst people whose houses have been flooded; either once or repeatedly. As such the paper is appropriately structured around this aim and the findings lined to the study's objectives.

I have two comments that the authors may wish to address:

Firstly, the background recognises prior research by the group into 12month and 2 year mental health issues following flooding. It is not entirely clear how this six months follow up study differentiates from these and from studies undertaken immediately after the flooding. The research question is does mental health sequelae following flooding change over time; either in severity or incidence.

Author response: As we mention in the Introduction section, Cumbria is an area that has been repeatedly affected by flooding. The data we collected from Cumbria therefore provided a unique opportunity to examine the impact of repeat flooding on mental health and health-related quality of life. A topic on which there has been limited research to date.

This study is different to our prior research, for several key reasons:

- The present study examines the impact of (repeat) flooding on health-related quality of life (as well as mental health outcomes). The previous research focuses solely on mental health outcomes. This is stated in the Introduction section (2nd paragraph) where we discuss the previous research "...The impact of flooding on health-related quality of life (HRQoL) has not yet been reported on."
- As mentioned, the present study specifically examines the association between repeat flooding and mental health/ health-related quality of life whereas the previous studies don't present any data on repeat flooding.
- The present study was conducted in Cumbria (in the north of England) whereas the previous studies were conducted in a much more affluent area of Southern England. We have clarified that the previously published data is for Southern England (Introduction, 2nd paragraph) "Data obtained one year after flooding in southern England showed...". The Objectives and Methods already clearly specify that the present study is conducted in the Cumbria region.

Secondly this point prevalence survey approach does not provide an enriched understanding of the actors that may be influencing the associations. Similar studies elsewhere involving qualitative analysis suggest that the post flooding management may be a significant factors along with the reality of risk exposure. Put simply, people can handle the flooding but have difficulty fighting with the insurance companies over the compensation. Perhaps the authors may wish to discuss the possible pathophysiological rationale of their observations in a little more depth.

Author response:

In the Introduction section (3rd paragraph) we do discuss the wide range of factors, including secondary stressors, that are known to influence mental health outcomes in the aftermath of flooding. It does not seem appropriate to repeat this within the Discussion section. However, in response to the authors comment we do now specifically mention “*difficulties with insurance or compensation*” and have added an appropriate reference for this:

“Several factors are known to be associated with the likelihood of experiencing adverse mental health outcomes post-flooding. These include socio-demographic characteristics such as age, gender, socio-economic and education statuses, and factors related to the nature of flooding e.g. depth and duration.[12] Secondary stressors, including problems with relationships, loss of sentimental items and difficulties with insurance or compensation have been associated with poor psychological outcomes.[13, 14] Meanwhile, societal-level factors such as social cohesion and support, as well as individual-level factors such as positive/proactive coping strategies can be protective against the development of mental ill health in the aftermath of flooding.[2, 14] Although there is less published research on the association between flooding and wellbeing/quality of life, the available evidence indicates that effects similarly occur through a complex set of inter-related pathways.[15]”

Of note, in the Discussion section we discuss in some depth, including the pathophysiological rationale for our main finding that people exposed to repeat flooding events do not appear to be at higher risk of poor mental health outcome (except possibly depression) compared to those exposed to a single flooding event.

“Repeat flooding may impact on mental health in a variety of ways, though the evidence is inconclusive.[14, 17, 18] At the individual level, experiencing a repeat episode of flooding may trigger memories of previous flooding episodes, intensifying the distress experienced.[17] Those who have been previously flooded may also experience anticipatory anxiety and stress about the prospect of future flooding.[31,32] Conversely, people’s individual and collective coping capacity may be increased, and they may display greater resilience to future flooding events.[33] There is also evidence to suggest that previous exposure to disasters may have an ‘inoculation’ effect, particularly among older adults (the median age of our study population was 62 years).[19, 34-35] Meanwhile, factors acting at the community level may mitigate the risks of adverse consequences of future floods through the development or strengthening of community networks and increased social cohesion.[15,31,36] Finally, past flood events may have resulted in improvements to infrastructure, flood defences and response systems, reducing the extent of subsequent floods and the impact on people’s lives.”

Reviewer: 2

Reviewer Name: Lennart Reifels

Institution and Country: Centre for Mental Health, Melbourne School of Population and Global Health, The University of Melbourne, Australia Please state any competing interests or state ‘None declared’:
None declared

Please leave your comments for the authors below

The authors have satisfactorily addressed all feedback points raised by this reviewer.

Minor remaining typos can likely be addressed during copy-editing:

(p.32, line): "The lack of association found [in] our study"

(p.33, line 13): "may have an ‘inoculation’ effect[s]"

(p.32, second paragraph): Since 'therapeutic interventions' are not usually indicated in the immediate aftermath of disasters, I would recommend to refer to Psychological First Aid as a "psychosocial support" strategy.

Author response: thank you for picking up on these minor typos. We have now corrected them. We have also replaced 'therapeutic interventions' with 'psychosocial support'.

VERSION 3 – REVIEW

REVIEWER	Gerry FitzGerald Queensland University of Technology, Queensland Australia
REVIEW RETURNED	09-Sep-2019

GENERAL COMMENTS	Thank you for responding to prior minor matters. This is an important contribution to understanding the longer term health consequences of disasters. I will be interested to see the longer term outcomes and the factors that influence those outcomes as it is these factors that authorities may be able to attend to.
--